# FDI and International Knowledge Diffusion: An Examination of the Evolution of Comparative Advantage

**Zhaobin Fan [1],\* , Hui Li [2] and Lin Pan [1]**

[1]   International Business School, Jinan University, 601 Huangpu W Ave, Guangzhou 510632, China; panlin@stu2015.jnu.edu.cn
[2]   School of Mathematics, University of Birmingham, Ring Rd W, Birmingham B15 2TS, UK; h.li.4@bham.ac.uk
\*   Correspondence: tfanzb@jnu.edu.cn

**Abstract:** This paper investigates the role of FDI in international knowledge diffusion with a focus on the evolution of the comparative advantage of FDI-host and FDI-source countries. We use data on 99 countries, which leads to 876 country pairs, over the 2001 to 2012 period. Spatial autoregressive (SAR) models are used to investigate the impact of bilateral FDI on the similarity of the comparative advantage between the host and source countries. Empirical results show that the effect of the bilateral FDI on the evolution of the comparative advantage of the host and source countries is statistically significant. Specifically, the larger the scale of bilateral FDI, the more similar the comparative advantage between the host and source countries becomes. We also find that the impact of FDI on international knowledge diffusion is heterogeneous across country pairs and this effect varies across the development gap between the source and host countries. In the case of countries that are not very different in terms of their level of economic development, bilateral FDI has a relatively more significant effect on the similarity of the comparative advantage between countries. Moreover, we find that the similarity of the comparative advantage is spatially correlated, and FDI linkages between country pairs strengthen the spatial correlations.

**Keywords:** FDI; knowledge diffusion; comparative advantage; spatial autoregressive model

---

## 1. Introduction

Globalization has had a far-reaching impact on trade and living standards over the past few decades, while foreign direct investment (FDI) flows have multiplied as well. From 1980 to 2016, FDI outflows increased by 26 times worldwide. For developed countries, the FDI outflows in 2016 were 139 times more than they were in 1980, and the FDI inflows were 86 times more. Whereas for developing countries, both the FDI outflows and inflows in 2016 were about 20 times more than they were in 1980. Moreover, between 1980 and 2016, the growth rate of FDI was higher than that of trade and gross domestic product (GDP) [1], the annual growth rate of FDI outflows reached 14%, which is twofold the annual growth rate of exports and threefold the annual growth rate of GDP.

FDI is not only a critical driver of global sustainable economic growth, but also plays a vital role in narrowing the economic development gap between the North and South. FDI is the largest and most stable source of external funding for developing countries, and, more importantly, it is also a significant source of advanced technology, which is of great importance to the structural transformation and upgrading of developing countries. Since Hymer [2], FDI is widely regarded as a "package" that involves a flow of capital, management, and knowledge, rather than just the international asset transactions. Moreover, "knowledge assets" is believed to be the core advantage of multinational

corporations [3], which provides a foundation for examining the role of FDI in international knowledge transfer and diffusion.

FDI-related knowledge diffusion includes horizontal diffusion and vertical diffusion [4]. The former refers to the knowledge diffusion generated by multinational corporations through the linkages with local intra-sector firms, and the latter refers to the diffusion of knowledge generated by multinational corporations through contacts with local inter-sector firms, including upstream and downstream firms.

Caves [5] pioneered the study of horizontal knowledge diffusion of FDI. Using the cross-sectional data of the manufacturing industry in Canada and Australia in 1966, he found that the productivity of local firms is positively associated with the scale of FDI, and thus he concluded that multinational corporations may have a knowledge diffusion effect on local competitors in the host market. This finding provides early evidence for the role of FDI in international knowledge diffusion. Since then, a large number of follow-up studies have drawn similar conclusions [6–18]. A few also found out that FDI has no significant or even negative impacts on the productivity of local firms due to its "crowding-out effect" [19–25].

Meanwhile, Lall [26] found that when multinational companies operate in the host countries, they not only have linkages with local rivals in the same sector, but also have linkages with upstream and downstream firms in different sectors. These backward and forward linkages may become an important channel for international knowledge diffusion of FDI. Many empirical studies have provided evidence for vertical knowledge diffusion effects through forward and backward linkages with local firms [23,24,27–32]. However, some studies also found that multinational companies have a negative impact on the productivity of local firms through vertical linkages [33,34].

In most recent studies, scholars have investigated the nonlinear effects of FDI on knowledge diffusion. Spencer [35] argued that FDI has a U-shaped effect on the productivity of local firms. Specifically, in the short term, due to the "crowding-out effect" from fierce competition, FDI negatively affects the productivity of local firms. In the long run, FDI will eventually lead to a positive impact on local companies through horizontal spillovers. Based on the data of China's manufacturing sector over the period of 2006–2009, Li and Tanna's [36] study provides evidence for a U-shaped effect of FDI on local firms at both horizontal and vertical linkage levels. On the contrary, using the data of manufacturing firms in China during 1998–2007, Xiao and Park [37] found that the effect of FDI on local firms' productivity exhibits an inverted U-shape instead. They argued that FDI has a positive impact in the short-term due to knowledge diffusion effects, but with the weakening of the knowledge diffusion over time, the "crowding-out effect" will finally dominate and make the net effect negative.

While a considerable amount of literature has produced a broad range of findings on the knowledge diffusion effect of FDI, most studies on FDI knowledge diffusion have focused on the perspective of host countries, and paid little attention to the impact of FDI on the relative competitive position between the FDI-host and FDI-source countries owing to knowledge diffusion. However, in the sustainability analysis of the impact of FDI on the convergence between the North and South, the perspective of source countries becomes equally important as the knowledge diffusion of FDI may weaken the competitive position of source countries. Moreover, previous studies have traditionally intended to capture the attribute of the knowledge diffusion effect based on the analysis of local productivity, and some recent studies have alternatively used patent applications, patent counts, or patent citations as measures of the knowledge diffusion effect of FDI [38]. While these measures provide an insightful assessment regarding the extent of the knowledge diffusion effect of FDI, to fully explore the effect of FDI on international knowledge diffusion, it is essential to account for other measures reflecting the relative competitive position between the FDI-host and FDI-source countries. Since the international knowledge diffusion can affect the comparative advantages of both source and host countries, this paper aims to study the role of FDI in knowledge diffusion by paying special attention to the evolution of the comparative advantages between the host and source countries.

Our study can provide new evidence for the role of FDI in international knowledge diffusion, and help understand the effect of FDI on the world distribution of income.

The remainder of this paper is organized as follows. Section 2 develops the hypotheses on the role of FDI in international knowledge diffusion from the viewpoint of the evolution of comparative advantages. Section 3 discusses the variable measurements and estimation specifications. Section 4 implements the spatial autoregressive (SAR) model to test the hypotheses. Section 5 summarizes the main conclusions, policy implications, and limitations.

## 2. Hypothesis Development

### 2.1. The General Effect of FDI on the Pattern of Comparative Advantage

The form of spatial diffusion of knowledge is different with the type of knowledge. Some knowledge is explicit and thus can be transmitted in an indirect way, such as international trade of goods and services. Some knowledge is disembodied and tacit, however, and can be transferred only in the form of more direct contacts [39]. Since tacit knowledge plays a significant role in the economic growth and the distribution of income, the international diffusion of tacit knowledge has long been of central interest for scholars and policymakers.

Since Hymer [2] pointed out the significant difference between foreign indirect investment and foreign direct investment (FDI), the role of FDI in the international diffusion of tacit knowledge has been increasingly recognized. First of all, unlike foreign indirect investment, FDI is not only the transnational flows of physical capital, but also the flows of knowledge, human capital, and management. Thus, it can be a carrier of tacit knowledge diffusion between the source and host countries. Moreover, compared with local firms in the host countries, multinational corporations have the advantage of "knowledge assets", which is the core advantage of multinational corporations, so they can generate knowledge diffusion. There are two basic types of international knowledge diffusion of FDI: Horizontal diffusion and vertical diffusion.

Horizontal knowledge diffusion of FDI occurs within the sector mainly through the demonstration effect, workers' mobility, and competition effect [4]. The demonstration effect suggests that local firms can learn by imitating the operations of multinational companies in the host market. Workers' mobility means that when the local workers employed by multinational companies move to local firms or start their own business, technological and managerial expertise can flow from multinational companies to local firms. The competition effect occurs when the entry of multinational companies forces local firms to improve their productivity through technological innovation or other measures.

Compared with horizontal diffusion, vertical diffusion of FDI mainly occurs through backward and forward linkages [26,28]. Backward linkages refer to the linkages of multinational companies with local upstream firms. In order to reduce the operation costs in host countries, multinational corporations usually intend to cooperate with local suppliers of intermediate inputs, and voluntarily provide technical guidance and training to the suppliers, such as product design, quality control, and inventory management, thus generating knowledge diffusion through backward linkages. Forward linkages refer to the linkages between multinational companies and local downstream firms. Local customers of multinational companies can benefit from spillovers embodied in products and processes. Furthermore, to promote the marketing of their products, multinational companies are willing to provide some post-sale technological services to the local customers (downstream firms), thus generating knowledge diffusions through forward linkages.

As FDI plays a significant role in international knowledge diffusion, the following question is how this effect will influence a country's trade pattern. According to the Ricardian framework of the comparative advantage model, the pattern of trade reflects the differences in productivity between countries. Each country exports the products in which it is relatively more productive and imports the products in which it is relatively less productive. Ceteris paribus, if a country relatively increases its productivity in a product, it will be more likely to become a potential exporter of that product.

The international diffusion of knowledge can affect the differences in productivity between countries, thus affecting the comparative advantage and trade pattern across countries. Considering the role of FDI in international knowledge diffusion, two countries with the linkages of FDI should have similar patterns of comparative advantage, and, therefore, will be exporting similar baskets of goods and services. Based on this, against no effect, we propose the following hypothesis:

**Hypothesis 1 (H1).** *Bilateral FDI affects the similarity of the comparative advantage between the host and source countries. The larger the bilateral FDI, the more similar the comparative advantage between the host and source countries.*

*2.2. The Heterogeneous Effect of FDI on the Pattern of Comparative Advantage*

While a large number of studies provided evidence for the role of FDI in knowledge diffusion, many studies also illustrated that the significance of this effect is determined by the absorptive capacity of host countries [21,24,30,37,40], which suggests that the effect of FDI on knowledge diffusion is heterogeneous across the country pairs. The absorptive capacity is defined as a dynamic capability pertaining to knowledge creation and utilization that enhances a firm's ability to gain and sustain a competitive advantage [41]. According to the theories of absorptive capacity, regardless of whether the knowledge diffusion of multinational companies is voluntary or not, due to the stickiness of knowledge diffusion, the external knowledge will not be easily converted into a local firm's internal knowledge. The extent of knowledge diffusion depends on the absorptive capacity of host countries.

The ability of the host country to absorb FDI's knowledge spillovers is determined by several factors. The first is the technological gap between the source and host countries. The existence of the technological gap is a prerequisite for knowledge diffusion, but the technological gap can also become an obstacle to knowledge absorption [10]. The greater the technological gap between the source and host countries, the more difficult it is for the host countries to absorb the knowledge spillovers of FDI [42]. Another factor is the accumulation of human capital in the host countries. Teece [43] found that even if the technological gap between the source and host countries is large, as long as human capital flourishes in the host countries, FDI can be still helpful for the technological progress of local firms. Moreover, previous research also found out that there is a human capital "threshold effect" in the knowledge diffusion of FDI: The host countries can only benefit from the knowledge diffusion of FDI when the accumulation of human capital reaches a certain level [9,44]. Moreover, the absorptive capacity is also associated with market maturity of the host countries. To maintain their market share in the host countries, the multinational corporations will usually use their monopoly power to prevent the local firms from absorbing the knowledge spillovers of FDI. The extent of the dominance of multinational corporations depends on the market maturity of the host countries. The better the host country's market mechanism performs, the less likely the multinational corporations are to monopolize the host countries' market. Therefore, the market maturity of the host countries will affect whether local firms can benefit from the multinational corporations [45]. The last factor is the difference in social background between the source and host countries. Some knowledge is usually rooted in the social context in which they emerged, such as the legal system, education system, cultural background, and labor market rules. It is usually not easy for the host countries to absorb FDI's knowledge spillovers unless they have a similar social background with the source countries [46].

Since knowledge diffusion of FDI depends on the absorptive capacity of the host countries, the effects of FDI on the similarity of comparative advantages can vary across the country pairs with different development gaps. For a long time, in the "South-North" FDI flows, the source of FDI was mainly the developed countries [1], hence the technological gap and social background between the host and source countries were relatively large. On the other hand, most developing countries have limited accumulation of human capital and often experience a significant loss of human capital. In addition, developing countries are usually less marketized and their markets are more likely to be

monopolized by multinational corporations. Therefore, if FDI occurs between "South-North" countries, the knowledge diffusion effect of FDI will be relatively weak, and the effects of bilateral FDI on the similarity of the comparative advantage will be less significant. However, if FDI occurs between "South-South" countries, two parties are more likely to have a smaller technological gap and exhibit a similar social context. Moreover, the narrower the technological gap between two parties, the lower the requirements for the accumulation of human capital and market maturity. Therefore, the effects of bilateral FDI on the similarity of comparative advantages between the "South-South" countries will be more significant. Finally, if FDI occurs between the "North-North" countries, the technological gap and social background difference between the host and source countries are relatively small. On the other hand, the accumulation of human capital and market maturity in both parties is relatively high. Thus, we would expect that the knowledge diffusion effect of FDI to be more significant in the "North-North" country pairs. Based on the above discussion, we propose the following theoretical hypothesis:

**Hypothesis 2 (H2).** *The effects of FDI on the similarity of comparative advantages are heterogeneous across the country pairs. The "North-North" and "South-South" FDI have more significant effects on the similarity of comparative advantages between the host and source countries than "South-North" FDI.*

### 2.3. FDI and the Spatial Correlation of the Pattern of Comparative Advantage

Spatial correlation is frequently considered in the fields that observations from nearby locations often share more common attributes than would be expected on a random basis, and it is also often highlighted in the relevant research on the regional economics [47]. It is essential to investigate the spatial correlation of comparative advantage, because if the pattern of comparative advantage does not spill over, the FDI location in one country may endanger the comparative position of adjacent areas, which may discourage regional corporations in attracting foreign capital inflows.

The pattern of comparative advantage can be spatially correlated between nearby countries for at least three reasons. First, adjacent areas are more likely to have similar resource endowments, and then develop a similar pattern of comparative advantage. Second, the economic policies of neighbors can influence the relative competitive position, so if neighboring countries alter their policies, other countries may feel competitive pressure to implement similar policies [48,49], which also can lead to a similar pattern of comparative advantage between neighboring countries. Finally, neighboring countries can more easily share knowledge. Due to relatively low mobility costs, adjacent areas are more likely to incur bilateral flows of goods and production factors, thereby having a higher spatial correlation of comparative advantage. Some studies have found that the diffusion of knowledge tends to decay with geographical distance, where they measured the knowledge diffusion by using patent citations [50], research and development (R&D) output [51,52], or total factor productivity [53]. In a recent study, Bahar et al. [54] investigated the impacts of geographical distance on knowledge diffusion by examining the evolution of comparative advantage, and found that neighboring countries share more knowledge, and have a higher similarity of comparative advantages.

Concerning the above spatial correlation, we expect that the similarity of comparative advantages between a country pair should be correlated to that of another country pair, and the spatial correlation between country pairs should be negatively associated with the average geographic distance between country pairs. In addition, since FDI is an essential channel for international knowledge diffusion, we expect that the spatial correlation of the comparative advantage similarity will be influenced by FDI linkages between country pairs. In particular, the spatial correlation of the comparative advantage similarity should be more significant for country pairs with FDI linkages. Therefore, we propose the following hypothesis:

**Hypothesis 3 (H3).** *The FDI linkages between the country pairs can strengthen the spatial correlation of the similarity of comparative advantages. In other words, the country pairs with FDI linkages have more significant spatial correlations in the similarity of comparative advantages.*

## 3. Estimation Specifications

### 3.1. Measurement of the Similarity of Comparative Advantages

Following Bahar et al.'s [54] study, we used the revealed comparative advantage (RCA) to measure the comparative advantage, and also employed the correlation coefficient of comparative advantage between two countries to measure their similarity of comparative advantage. That is:

$$RCA_{c,p} = \frac{X_{c,p} / \sum_p X_{c,p}}{\sum_c X_{c,p} / \sum_c \sum_p X_{c,p}}, \tag{1}$$

where $RCA_{c,p}$ is the revealed comparative advantage of country $c$ in product $p$, and $X_{c,p}$ denotes the export of country $c$ in product $p$.

$$S_{c,c'} = \frac{\sum_p \left( r_{c,p} - \bar{r}_c \right) \sum_p \left( r_{c',p} - \bar{r}_{c'} \right)}{\sqrt{\sum_p \left( r_{c,p} - \bar{r}_c \right)^2 \left( r_{c',p} - \bar{r}_{c'} \right)^2}}, \tag{2}$$

where $S_{c,c'}$ is the similarity of comparative advantages between two countries, $c$ and $c\prime$, and $r_{c,p} = ln(RCA_{c,p})$. $\bar{r}_c$ is the average of $r_{c,p}$ over all products in country $c$. The reason for choosing the log form is to prevent the correlation from being affected by the few products with a very high RCA.

The data used to calculate the similarity of comparative advantage was sourced from the World Integrated Trade Solution (WITS) database of the United Nations Conference on Trade and Development (UNCTAD). WITS data includes information on the revealed comparative advantage of 30 product categories, which include Animal, Vegetable, Food Products, Minerals, Fuels, Chemicals, Plastic or rubber, Food Products, Hides and skins, Wood, Textiles and Clothing, Footwear, Stone and glass, Metals, Mach and Elec, Transportation, Miscellaneous, Agricultural Raw materials, Chemicals, Food, Fuel, Manufactures, Ores and metals, Textiles, Machinery and Transport Equipment, Raw materials, Intermediate goods, Consumer goods, Capital goods, Chemical.

### 3.2. Specification of the Econometric Model

To account for the spatial correlations between the country pairs, we used the spatial autoregressive (SAR) model to analyze the effects of FDI on international knowledge diffusion. The SAR model is widely applied to cases where outcomes of a spatial unit at one location depend on those of its neighbors. The estimation model for testing the hypothesis, H1 and H3, is as follows:

$$
\begin{aligned}
S_{c,c',t} ={}& \alpha + \beta FDI_{c,c',t} + \rho(W \times S_{q,q',t}) + \lambda Distance_{c,c'} + \phi_1 Border_{c,c'} + \phi_2 Language_{c,c'} \\
& + \phi_3 Colonizer_{c,c'} + \gamma_1 DGDPPPC_{c,c',t} + \gamma_2 DPOP_{c,c',t} + \gamma_3 DPCPW_{c,c',t} \\
& + \gamma_4 DHCPW_{c,c',t} + \gamma_5 DLPW_{c,c',t} + \varepsilon_{c,c',t}
\end{aligned}
\tag{3}
$$

where the dependent variable, $S_{c,c',t}$, is the similarity of comparative advantages between two countries, $c$ and $c\prime$, in period $t$, which is in its logarithmic form. $\varepsilon$ is the error term following log-normal distribution.

$FDI_{c,c',t}$ represents the sum of bilateral FDI between country $c$ and $c\prime$ at time $t$, which is also in its logarithmic form. We investigated the effects of FDI flow and stock, respectively. If the coefficient, $\beta$, is positive, then the data support hypothesis H1. The data on bilateral FDI were sourced from the UNCTAD database.

$S_{q,q',t}$ is the similarity of the comparative advantage of any other country pairs at time $t$. W is the weight matrix that accounts for the spatial relationships (dependencies) among the spatial data. $\rho$ is the spatial autoregression (autocorrelation) parameter. $\rho(W \times S_{q,q',t})$ is the spatial autocorrelation term that models the strength of the spatial dependencies among the elements of the dependent variable. Based on the information of the geographical distance, we constructed two spatial weight matrices: W1 and W2. We let W2 contain information on FDI linkages between country pairs, which was used to test the hypothesis, H3.

$Distance_{c,c'}$ represents the logarithmic transformation of the geographical distance between the biggest city (measured in population) of two countries, $c$ and $c'$. The data on the geographical distance was derived from the Center for Research and Expertise on the World Economy (CEPII) database. According to the study by Bahar et al. [54], the similarity of comparative advantages will decay with geographical distance between countries.

We constructed three dummy variables to control the geographical and cultural commonality between countries. Specifically, $Border_{c,c'}$ is a dummy variable indicating whether country $c$ and $c'$ share a border. If two countries are contiguous, it takes the value of 1, or 0 otherwise. $Language_{c,c'}$ is also a dummy variable. It equals one if country $c$ and $c'$ have the same official language. $Colonizer_{c,c'}$ measures whether country $c$ and $c'$ have a colony–colonizer relationship in history after 1945. If they do, then it takes the value of one, otherwise zero. We expect that the existence of commonalities should be positively associated with the similarity of the comparative advantage between two countries. The data on the above three dummy variables was compiled from the CEPII database.

$DGDPPC_{c,c',t}$ represents the absolute value of differential in the logarithmic values of the real GDP per capita between two countries in period $t$. It was used to measure the gap of economic development between two countries. According to the trade theory of preference similarity [55], countries with a similar income level have a similar trade pattern. Therefore, the gap of the development level should be negatively associated with the similarity of the comparative advantage. The data in real GDP per capita (2010 constant dollars) were from the World Bank database.

$DPOP_{c,c',t}$ is the absolute value of differential in the logarithmic values of the population between two countries, $c$ and $c'$, in period $t$, which was used to measure the similarity of the market size between the two countries. According to the new trade theory, market size is also an important determinant for the pattern of comparative advantage. The closeness of the market size is expected to be positively associated with the similarity of the comparative advantage. The data on the population was derived from the World Bank database.

$DPCPW_{c,c',t}$ represents the absolute value of differential in the capital-labor ratio, which takes the logarithmic form between $c$ and $c'$ in period $t$. $DHCPW_{c,c',t}$ is the absolute value of the differential in the average education level in the logarithmic form between two countries in period $t$. $DLPW_{c,c',t}$ is the absolute value of differential in the land-labor ratio in the logarithmic form between countries c and $c'$ in period $t$. These three variables were used to control the differentials in resource endowments between two countries. According to the neoclassical trade theory, the lower the difference in resource endowments, the higher the similarity of the comparative advantage.

In order to test to whether FDI's knowledge diffusion effect is heterogeneous across country pairs, which is the hypothesis H2, we introduced an interaction term between FDI and a dummy variable, $B_{c,c'} \times FDI_{c,c',t}$, into the model, where $B_{c,c'}$ indicates whether a country pair has a similar development level. If both countries, $c$ and $c'$, are developed countries or developing countries, then $B_{c,c'} = 1$, otherwise $B_{c,c'} = 0$. That is:

$$
\begin{aligned}
S_{c,c',t} &= \alpha + \beta(B_{c,c'} \times FDI_{c,c',t}) + \rho(W \times S_{q,q',t}) + \lambda Distance_{c,c'} + \phi_1 Border_{c,c'} + \phi_2 Language_{c,c'} \\
&\quad + \phi_3 Colonizer_{c,c'} + \gamma_1 DGDPPC_{c,c',t} + \gamma_2 DPOP_{c,c',t} + \gamma_3 DPCPW_{c,c',t} + \gamma_4 DHCPW_{c,c',t} \\
&\quad + \gamma_5 DLPW_{c,c',t} + \varepsilon_{c,c',t}
\end{aligned} \tag{4}
$$

where if the parameter of the interaction term, $\beta$, is significantly positive, then the bilateral FDI are more likely to have a positive effect on the similarity of comparative advantages between two countries with a similar development level. In other words, two countries with a similar development level absorb each other's FDI knowledge spillovers more easily. The identification of developed and developing countries follows the standard of the International Monetary Fund (IMF), where 39 countries are classified as developed countries.

### 3.3. Spatial Weight Matrix

Since the sample is an unbalanced panel data, we constructed 12 small spatial sub-matrices for 12 periods of 2001–2012, and then combined 12 small sub-matrices into an aggregated diagonal matrix. $w_{ijt}$ is an item of a sub-matrix for period $t$, where $i$ represents the country pair of $c$ and $cı$, and $j$ denotes the country pair of $q$ and $qı$, then:

$$w_{ijt} = \frac{1}{average(d_{c,q} + d_{c,qı} + d_{cı,q} + d_{cı,qı})} \tag{5}$$

where, $d_{c,qı}$ denotes the geographical distance between $c$ and $q$, $d_{c,qı}$ is the geographical distance between $q$ and $qı$, etc. For example, if $c$ and $cı$ denote Algeria and France, respectively, and $q$ denotes the United States, then $d_{c,c} = 0$, $d_{c,q} = 6471.879$, $d_{cı,q} = 1340.39$, and $d_{cı,q} = 5838.157$. Moreover, if $i$ denotes the country pair of Algeria and France, and $j$ denotes the country pair of Algeria and the United States, then according to Equation (5), $w_{ijt} = 0.00028$. Unlike the neighborhood matrix formation, which only considers whether two countries are adjacent, the spatial weight matrix in this paper contains more geographical information. Based on the discussion above, the spatial weight sub-matrix for period $t$, $W_t$, is constructed as follows:

$$W_t = \begin{pmatrix} 0 & \cdots & w_{n1t} \\ \vdots & \ddots & \vdots \\ w_{1nt} & \cdots & 0 \end{pmatrix}, \tag{6}$$

where the geographical distance between the country pair $i$ and itself is zero. Using 12 small sub-matrices for 12 periods from 2001 to 2012, we then aggregated them into a final diagonal matrix, W, as follows:

$$W = \begin{bmatrix} W_{2001} & \cdots & O \\ \vdots & \ddots & \vdots \\ O & \cdots & W_{2012} \end{bmatrix}, \tag{7}$$

where the elements along the diagonal line are composed of 12 small sub-matrices, and the off-diagonal is filled by sub-matrices containing only zeroes. The elements of the spatial matrix, W, were standardized.

In order to test the hypothesis, H3, based on matrix (7), we needed to set up another spatial matrix that incorporated the bilateral FDI information between country pairs. If any country in the country pair, $i$, is also in the country pair, $j$, we used formula (5) to calculate the spatial weight between the country pair, $i$ and $j$. However, if none of the countries in the pair, $i$, are included in the pair, $j$, the spatial weight between the country pair, $i$ and $j$, is set to zero. For example, for the country pair of China-US vs China-Japan, we used the geographical distance between China and the United States, Japan and China, and Japan and the United States to calculate the spatial weight between the two country pairs. For the country pair of China-US vs. Japan-Korea, the spatial weight between the two country pairs was set to zero. This spatial matrix contains the information on bilateral FDI linkages, which can be used to investigate the spatial spillover effects of FDI's knowledge diffusion.

We defined W1 as the spatial matrix without considering bilateral FDI linkages, and W2 as the spatial matrix incorporating information on bilateral FDI between country pairs. Figure 1 is the image

of the matrix, W1 and W2, simulated using Matlab, where both W1 and W2 are a diagonal matrix of 5282 × 5282. The number of observations in 2001 was the smallest, which contained 309 observations, while the sample size in 2007 being the biggest, which was composed of 546 observations.

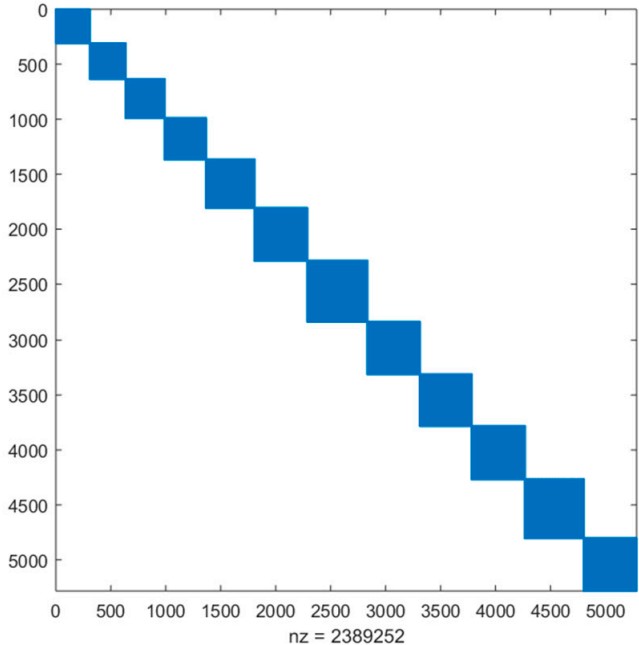

**Figure 1.** The image of the spatial weight matrix

We implemented Moran's I index to test the spatial autocorrelation against the null hypothesis that the data is randomly distributed. Results are presented in Table 1, which reject the null hypothesis at the 1% level of significance, suggesting that the dependent variable is spatially correlated in the sample.

**Table 1.** The results of the Moran test.

| | | |
|---|---|---|
| W1 | Moran I | 0.0440 *** |
| | Moran I-statistic | 0.0237 *** |
| W2 | Moran I | 0.1990 *** |
| | Moran I-statistic | 0.1280 *** |

Notes: *** denotes the significance level of 1%.

*3.4. Data and Descriptive Statistics*

Mainly due to the data limitation of bilateral FDI (stock and flow), the data set used in our study consists of 99 countries and 876 country pairs over the period of 2001–2012. It includes 36 developed countries and 446 country pairs with similar economic development levels.

The descriptive statistics are presented in Table 2. As can be seen, the correlations between most of the independent variables are low. The mean for all variables is positive and falls within the interval [0, 2]. Specifically, the mean of comparative advantage similarity is 0.17, which implies that for most countries, their comparative advantages are positively correlated with each other. In addition, the standard deviation of all variables is small, which is helpful to exclude the impacts of outliers.

For the similarity of the comparative advantage, the minimum value is -0.81, which takes place between Algeria and the United States, and the maximum is 0.91 (between Austria and Slovenia). The country pair with the largest FDI flows was Luxembourg and the United States in 2011, and the smallest flow comes from Czech and Lithuania in 2008. The country pair with the largest FDI stock

was the United Kingdom and the United States (2012), and the one with the smallest was Estonia and Turkey (2006). There are 68 country pairs sharing a common border, and 48 country pairs having the colony–colonizer relationship and 105 pairs of countries using the common official language.

**Table 2.** Coefficients of correlation and descriptive statistics.

| | S | FDI-Stock | FDI-Flows | Distcance | Border | Colonizer | Language | DGDPPC | DPOP | DPCPW | DHCPW | DLPW |
|---|---|---|---|---|---|---|---|---|---|---|---|---|
| S | 1 | 0.25 *** | 0.22 *** | −0.32 *** | 0.30 *** | 0.08 *** | 0.10 *** | −0.33 *** | 0.025 * | −0.30 *** | −0.26 *** | −0.04 *** |
| FDI-stock | | 1 | 0.56 *** | −0.10 *** | 0.18 *** | 0.16 *** | 0.19 *** | −0.20 *** | −0.06 *** | −0.22 *** | −0.11 *** | −0.06 *** |
| FDI-flows | | | 1 | −0.12 *** | 0.17 *** | 0.15 *** | 0.17 *** | −0.19 *** | −0.04 *** | −0.21 *** | −0.11 *** | −0.07 *** |
| Distance | | | | 1 | −0.45 *** | −0.05 *** | −0.01 | 0.23 *** | 0.14 *** | 0.24 *** | 0.24 *** | 0.16 *** |
| Border | | | | | 1 | 0.20 *** | 0.20 *** | −0.18 *** | −0.05 *** | −0.16 *** | −0.11 *** | −0.02 |
| Colonizer | | | | | | 1 | 0.38 *** | 0.02 * | −0.01 *** | 0.03 *** | 0.15 *** | −0.06 *** |
| Language | | | | | | | 1 | −0.01 | 0.03 *** | 0.01 | 0.16 *** | 0.04 *** |
| DGDPPC | | | | | | | | 1 | −0.04 *** | 0.95 *** | 0.55 *** | 0.01 |
| DPOP | | | | | | | | | 1 | −0.06 *** | −0.06 *** | 0.40 *** |
| DPCPW | | | | | | | | | | 1 | 0.55 *** | −0.02 * |
| DHCPW | | | | | | | | | | | 1 | 0.00 |
| DLPW | | | | | | | | | | | | 1 |
| Observations | 5282 | 5282 | 5282 | 5282 | 5282 | 5282 | 5282 | 5282 | 5282 | 5282 | 5282 | 5282 |
| Mean | 0.17 | 20.84 | 18.77 | 7.83 | 0.10 | 0.06 | 0.11 | 1.09 | 1.83 | 1.02 | 0.17 | 4.53 |
| S.D. | 0.01 | 0.04 | 0.03 | 0.02 | 0. | 0 | 0 | 0.01 | 0.02 | 0.01 | 0.01 | 0.05 |
| Minimum | −0.81 | 12.35 | 9.23 | 4.39 | 0 | 0 | 0 | 0.00002 | 0.0004 | 0.0004 | 0.00003 | 0.0011 |
| Maximum | 0.91 | 27.34 | 26.03 | 9.88 | 1 | 1 | 1 | 4.63 | 7.93 | 5.09 | 1.90 | 17.94 |

Notes: *** denotes the significance level of 1%, ** denotes the significance level of 5%, and * denotes the significance level of 10%.

## 4. Estimation Results

### 4.1. The General Effect of FDI on the Pattern of Comparative Advantage

In this part, based on the econometric model (I), this paper performed SAR regression to test the hypotheses H1 and H3. We also provide the OLS estimation results to compare with the results from the SAR regression. The FDI stock and flow were used as the explanatory variable, respectively.

Table 3 presents the regression results using the FDI stock as the explanatory variable. Column (1) shows the OLS regression results without incorporating the spatial autoregression term, while column (2) and (3) are the SAR regression results from including the spatial autoregression term. As shown in the OLS regression results, the coefficient of FDI stock is significantly positive at the level of 1%, and a 1% increase in the stock of bilateral FDI will improve the similarity of the trade pattern by 2%. This result supports the theoretical hypothesis, H1, that is, bilateral FDI will affect the pattern of the comparative advantage of the host and source countries, and the larger the bilateral FDI, the more similar the comparative advantages.

All control variables significantly affect the similarity of comparative advantage except for *Colonizer* (colonial relationship) and *DLWP* (land endowment). The coefficient of *Distance* (geographical distance) is negative, which is in line with the theoretical expectation that the international knowledge diffusion of FDI decays with geographical distance. For another geographical variable, *Border*, its coefficient is significantly positive, implying that two countries sharing a border are more likely to have the similar pattern of comparative advantage. The coefficient of the dummy variable, *language*, is positive, which confirms the prediction that speaking a common language can improve international knowledge diffusion. The gap of economic development, *DGDPPC*, has a negative impact on the similarity of trade pattern, which is consistent with the theoretical prediction. The differences in the market size and physical capital are positively associated with the similarity of the comparative advantages, which deviates from the theoretical expectation. As expected, the impact of the human capital differential on the similarity of the comparative advantage is significantly negative.

Column (2) of Table 3 shows the regression results of the SAR model using the spatial weight matrix, W1. As can be seen, after including the spatial autoregression term in the regression, the coefficient of the FDI stock is still significantly positive at the level of 1%, but it becomes smaller than the result from OLS regression. For the control variables, their signs and significance levels are quite similar to the results from OLS regression. The coefficient of the spatial autoregression term is positive and significant at the level of 1%. This result confirms that the similarity of the comparative

advantage is spatially correlated between country pairs, and this correlation decays with the average geographical distance between country pairs. Specifically, one unit of increase in the similarity of the trade pattern between other countries will cause the similarity of the observed country pair to increase by 0.463.

**Table 3.** The estimation results in the case of using the FDI stock as the explanatory variable.

| Variable | (1) | (2) | (3) |
|---|---|---|---|
| | **OLS** | **SAR (W1)** | **SAR (W2)** |
| *Constant* | 0.2683 *** (5.509) | 0.1147 * (1.859) | 0.0066 (0.142) |
| *FDI−stock* | 0.0202 *** (12.119) | 0.0197 *** (11.814) | 0.0166 *** (10.412) |
| *Distance* | −0.0593 *** (−13.604) | −0.0506 *** (−10.452) | −0.0347 *** (−8.314) |
| *Border* | 0.1487 *** (9.065) | 0.1547 *** (9.4186) | 0.1580 *** (10.059) |
| *Colonizer* | 0.0299 (1.586) | 0.0290 (0.0290) | 0.0335 * (1.859) |
| *Language* | 0.0508 *** (3.453) | 0.0524 *** (3.569) | 0.0372 ** (2.643) |
| *DGDPPC* | −0.1413 *** (−9.206) | −0.1431 *** (−9.3460) | −0.1364 *** (−9.282) |
| *DPOP* | 0.0151 *** (4.638) | 0.0150 *** (4.616) | 0.0120 *** (3.831) |
| *DPCPW* | 0.0841 *** (5.192) | 0.0859 *** (5.315) | 0.0774 *** (4.992) |
| *DHCPW* | −0.1852 *** (−7.076) | −0.1790 *** (−6.843) | −0.1531 *** (−6.111) |
| *DLPW* | −0.0014 (−1.123) | −0.0012 (−0.924) | −0.0003 (−0.284) |
| $W \times S$ | | 0.4630 *** (4.043) | 0.6580 *** (191.588) |
| Observations | 5282 | 5282 | 5282 |
| Adj.$R^2$ | 0.2313 | 0.2320 | 0.2254 |

Notes: *** denotes the significance level of 1%, ** denotes the significance level of 5%, and * denotes the significance level of 10%. Standard errors are reported in parentheses.

The regression results of the SAR model using the spatial weight matrix, W2, are reported in column (3). The coefficient of the FDI stock continues to be positive and significant at the level of 1%, which further supports the impact of FDI on the pattern of the comparative advantage. For the control variables, the estimation results are very similar to the case of the OLS regression and the SAR regression using the spatial weight matrix, W1. The coefficient of the spatial autoregression term is significantly positive at the level of 1%. Specifically, it is 0.658, which is greater than the value of 0.463 as in the case of estimation using the spatial weight matrix, W1. Thus, the result confirms the theoretical hypothesis, H3, that the FDI linkages between country pairs will strengthen the spatial correlation of the similarity of comparative advantages.

Table 4 reports the estimation results using the FDI flow as the explanatory variable. Similar to Table 3, we provide the estimation outcomes from OLS, SAR (W1), and SAR (W2). Across the three models, the coefficient of FDI flow is positive and significant at the 1% level, which again supports

the hypothesis, H1. The coefficient of the spatial autoregression term in both SAR (W1) and SAR (W2) models is significantly positive, which strongly indicates that the similarity of the comparative advantage is spatially correlated between country pairs. Moreover, the coefficient of the spatial auto-regression term in SAR (W2) is still larger than the case of the SAR (W1), which further confirms the hypothesis, H3. The signs and significance levels of other control variables are largely consistent from the results in Table 3. The main change is that the colonial relationship is no longer significant in the case of using the FDI flow as the explanatory variable.

**Table 4.** The estimation results in the case of using FDI flow as the explanatory variable.

| Variable | (1) | (2) | (3) |
|---|---|---|---|
| | OLS | SAR (W1) | SAR (W2) |
| Constant | 0.3741 *** | 0.2130 *** | 0.0988 ** |
| | (7.747) | (4.431) | (2.140) |
| FDI−flows | 0.0163 *** | 0.0158 *** | 0.0131 *** |
| | (9.312) | (9.041) | (7.807) |
| Distance | −0.0574 *** | −0.0485 *** | −0.0331 *** |
| | (−3.113) | (−11.106) | (−7.896) |
| Border | 0.1576 *** | 0.1638 *** | 0.1656 *** |
| | (9.579) | (9.980) | (10.524) |
| Colonizer | 0.0330 * | 0.0322 * | 0.0364 ** |
| | (1.742) | (1.703) | (2.011) |
| Language | 0.0594 *** | 0.0609 *** | 0.0445 *** |
| | (4.0231) | (4.1390) | (3.156) |
| DGDPPC | −0.1385 *** | −0.1404 *** | −0.1339 *** |
| | (−8.972) | (−9.125) | (−9.075) |
| DPOP | 0.0140 *** | 0.0139 *** | 0.0110 *** |
| | (4.259) | (4.243) | (3.501) |
| DPCPW | 0.0775 *** | 0.0795 *** | 0.0717 *** |
| | (4.764) | (4.897) | (4.605) |
| DHCPW | −0.1879 *** | −0.1813 *** | −0.1553 *** |
| | (−7.140) | (−6.910) | (−6.172) |
| DLPW | −0.0015 | −0.0012 | −0.0004 |
| | (−1.155) | (−0.953) | (−0.323) |
| $W \times S$ | | 0.4830 *** | 0.6610 *** |
| | | (52.665) | (193.053) |
| Observations | 5282 | 5282 | 5282 |
| Adj.$R^2$ | 0.2227 | 0.2235 | 0.2170 |

Notes: *** denotes the significance level of 1%, ** denotes the significance level of 5%, and * denotes the significance level of 10%. Standard errors are reported in parentheses.

### 4.2. The Heterogeneous Effect of FDI on the Pattern of Comparative Advantage

We employed the econometric model (II) to test the hypothesis, H2, regarding whether the impact of FDI on international knowledge diffusion is heterogeneous across country pairs due to the absorptive capability. We still used the FDI stock and flow as the explanatory variable, respectively. The SAR estimation results are shown in Table 5, and SAR(W1) and SAR(W2) were again used to represent two cases of spatial matrix.

Across the four SAR models, the coefficients of the interaction term of FDI and the dummy variable are all positive and significant at the 1% level, implying that compared with "North-South" FDI, "North-North" and "South-South" FDI are more likely to influence the similarity of the comparative advantage. This result is highly consistent with the hypothesis, H2. We also find that the coefficient of the interaction term of FDI in SAR (W2) is consistently lower than the result in the case of SAR (W1).

The empirical results in Table 5 suggest that the heterogeneity of FDI's effect on the similarity of country pairs is highly robust. In addition, as shown in Table 5, the coefficient of the spatial

autoregression term is significantly positive at the 1% level across all regressions, and it is consistently higher in the case of using W2 as the spatial weight, which further supports the hypothesis, H3.

**Table 5.** The heterogeneous effect of FDI on the pattern of the comparative advantage.

| Variable | FDI stock | | FDI flow | |
|---|---|---|---|---|
| | **(1)** | **(2)** | **(3)** | **(4)** |
| | **SAR (W1)** | **SAR (W2)** | **SAR (W1)** | **SAR (W2)** |
| *Constant* | 0.3805 *** | 0.2537 *** | 0.3818 *** | 0.2542 *** |
| | (10.121) | (7.021) | (10.137) | (7.020) |
| $B \times FDI-stock$ | 0.0040 *** | 0.0032 *** | | |
| | (7.632) | (6.308) | | |
| $B \times FDI-flow$ | | | 0.0044 *** | 0.0034 *** |
| | | | (7.451) | (6.118) |
| *Distance* | −0.0417 *** | −0.0283 *** | −0.0417 *** | −0.0282 *** |
| | (−9.374) | (−6.625) | (−9.342) | (−6.586) |
| *Border* | 0.1789 *** | 0.1776 *** | 0.1795 *** | 0.1781 *** |
| | (10.911) | (11.298) | (10.945) | (11.327) |
| *Colonizer* | 0.0594 *** | 0.0586 *** | 0.0583 *** | 0.0578 *** |
| | (3.138) | (3.232) | (3.083) | (3.184) |
| *Language* | 0.0609 *** | 0.0449 *** | 0.0613 *** | 0.0453 *** |
| | (4.116) | (3.169) | (4.147) | (3.194) |
| *DGDPPC* | −0.1044 *** | −0.1050 *** | −0.1053 *** | −0.1058 *** |
| | (−6.570) | (−6.890) | (−6.626) | (−6.946) |
| *DPOP* | 0.0115 *** | 0.0090 *** | 0.0114 *** | 0.0090 *** |
| | (3.488) | (2.865) | (3.459) | (2.841) |
| *DPCPW* | 0.0698 *** | 0.0633 *** | 0.0697 *** | 0.0631 *** |
| | (4.313) | (4.077) | (4.304) | (4.067) |
| *DHCPW* | −0.2151 *** | −0.1826 *** | −0.2141 *** | −0.1815 *** |
| | (−8.103) | (−7.176) | (−8.065) | (−7.135) |
| *DLPW* | −0.0018 | −0.0009 | −0.0018 | −0.0009 |
| | (−1.394) | (−0.720) | (−1.380) | (−0.705) |
| $W \times S$ | 0.5280 *** | 0.6640 *** | 0.5290 *** | 0.6670 *** |
| | (59.179) | (195.257) | (56.414) | (194.142) |
| Observations | 5282 | 5282 | 5282 | 5282 |
| Adj.R$^2$ | 0.2180 | 0.2117 | 0.2176 | 0.2183 |

Notes: *** denotes the significance level of 1%. Standard errors are reported in parentheses.

## 5. Discussion and Limitations

The impact of FDI on cross-border knowledge diffusion has long been the subject of intense debate among researchers. Most studies on FDI knowledge diffusion have adopted the perspective of host countries in an attempt to investigate how inward FDI sustains the growth of host economies, and paid little attention to the impact of FDI on the relative competitive position between the host and source countries. Unlike the existing literature, while examining the FDI knowledge diffusion, this paper focuses on the impact of FDI on the relative competitive position between the FDI-host and FDI-source countries by paying special attention to the evolution of the comparative advantage of both the source and host countries. Using a sample consisting of 99 countries, 876 country pairs, and 5282 observations during the period of 2001–2012, we implemented the spatial autoregressive model to estimate the impacts of bilateral FDI on the similarity of the comparative advantage between the host and source countries.

The empirical results show that the bilateral FDI is associated with the similarity of the comparative advantage between the source and host countries; specifically, the larger the bilateral FDI, the more similar the comparative advantages between any two countries will be. This result not only confirms the favorable role of FDI in global knowledge transfer and diffusion that is emphasized

by a large number of previous studies since Caves [5], but also suggests that FDI has an impact on the relative competitive position of the host and source countries; in particular, it can promote the convergence of the competitive position of the host and source countries.

Moreover, we find that the impact of bilateral FDI on international knowledge diffusion is heterogeneous across country pairs, and varies according to the development gap between the source and host countries. In particular, "North-North" and "South-South" FDI has a more significant impact on the similarity of the comparative advantages of country pairs than "South-North" FDI. This result supports the prediction that the significance of the knowledge transfer of FDI is associated with the absorptive capability of host countries, which has also been highlighted in previous relevant literature [21,24,30,37,40]. Furthermore, we find that the similarity of the comparative advantage is spatially correlated between country pairs and this correlation decays with the geographical distance between country pairs. This empirical result is in line with the previous finding that there is a spatial spillover effect of knowledge, and neighboring countries share more knowledge [54]. Finally, we find that FDI linkages between country pairs can strengthen the spatial correlations of comparative advantage similarity. In other words, the country pairs with FDI linkages have more significant spatial correlations in the similarity of comparative advantages. This result further confirms the favorable effect of FDI flows on international knowledge diffusion.

Our study provides some compelling implications for policymakers. First, FDI can indeed help the host countries to develop their comparative advantage in modern industries and promote the transformation and upgrading of economic structure. This implies that to achieve sustainable economic growth, developing countries are expected to take further steps to liberalize their domestic markets, and correspondingly attract more foreign capital inflows. Second, FDI can reduce the level of income inequality across country groups, especially by narrowing the income gap between the developed and developing countries. Therefore, in order to promote the convergence of income per capita across countries, policies need to be designed to encourage global FDI flows between developing and developed countries. Furthermore, considering the spatial correlations of FDI's knowledge diffusion, developing countries should promote more in-depth regional cooperation and create a sound business environment for FDI flows within the adjacent regions. Finally, against the background of the surge of anti-globalization in some developed countries, developing countries can benefit in strengthening the "South-South" FDI to tackle the problem of trade protectionism.

We close our discussion by addressing the following caveats and limitations, which can be extended in further research. First, the measurement of competitive advantage in our study does not capture trade with processing goods, because "what countries export may be very different from what they actually contribute to the production process" [56]. It will be interesting if future studies could distinguish general trade from processing trade when measuring the competitive advantage. Next, due to the data limitation, we do not distinguish horizontal FDI from vertical FDI; given the difference between horizontal and vertical knowledge diffusion, it would be very valuable if further research could investigate the respective impacts of the two types of FDI on the similarity of the comparative advantage between the source and host countries. Finally, because of the empirical strategy, we did not examine the impact of FDI on the dynamics of export similarity. Future studies could use a dynamic analysis to study how FID makes the host countries add a particular product to their export basket or improve their comparative advantage in a particular product.

**Author Contributions:** Z.F. developed the theoretical hypotheses and wrote the paper. H.L. contributed to the introduction and discussion. L.P. collected data and conducted the empirical analysis. All authors improved and revised the final manuscript.

**Funding:** This research was funded by the Fundamental Research Funds for the Central Universities of China, grant number 15jnlh007.

**Conflicts of Interest:** The authors declare no conflict of interest.

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
