# Peer review of "FDI and International Knowledge Diffusion: An Examination of the Evolution of Comparative Advantage"

_sustainability, doi:10.3390/su11030581_

Round 1
Reviewer 1 Report
The paper draws on a compelling topic - i.e., the role of FDI in international knowledge diffusion in relation to the evolution of comparative advantage between FDI-host and FDI-source countries. The advanced perspective is insightful whereas the methodology is accurate and well-grounded.
However, in order to improve their paper and to increase its scientific soundness, the authors may consider the following suggestions:
it would be beneficial if the conceptual / theoretical framework is updated with more topical references - in its current form, there are few scientific studies published after 2016 cited in the text; the inclusion of at least 4-5 new sources would provide the required freshness and relevance to the overall approach;
the relationship of the investigated topic with the issue of sustainable development should be better argued throughout the study in order to be more consistent with the scope of the journal;
the discussion of the results should include specific comments regarding how the current findings relate to the previous ones, explicitly mentioning the similarities and the differences between them;
the final section of the paper should also include the limitations of the study in terms of conceptual model, method, sample, context, etc. Also, suggesting further specific research avenues would be very useful as a potential guide for scholars interested in this topic.
I wish the authors best of luck in revising their manuscript!
Author Response
Dear reviewer,
Thank you very much for your comments concerning our manuscript entitled “FDI and International Knowledge Diffusion: An Examination of the Evolution of Comparative Advantage”. Those comments are all valuable and very helpful for revising and improving our paper. We have studied comments carefully and have modified the manuscript accordingly. Revised portion is clearly highlighted using red color in the latest manuscript. The main corrections in the paper and the responds to reviewer’s comments can be seen in the attach file.

Reviewer 2 Report
I congratulate the authors both for the research carried out and for the content of the paper. With respect to research, the methodology is appropriate, the scientific rigor is high, the results taking into account the goodness of fit of the model are reliable. In summary, the research presents high levels of scientific rigor.
On the other hand, the paper is easy to read, is correctly structured, each section contains the required information, the model is based on the theoretical framework that is correct, the methodology is correctly described including all the relevant information. The results are described in detail. The authors have written the conclusions taking into account the results obtained and have adequately stated for those who are valid these results; Theoretical implications and Managerial implications.
My only suggestion is to introduce the limitations of the study into the conclusions.
The authors must follow the layout guidelines of the journal. See guidelines regarding the bibliography incorporated in the text and in the references section.
I believe that the work in can be published in its current state.
Author Response

(The authors gave the same response as above.)

Reviewer 3 Report
This manuscript examines the effect of FDI on the similarity of comparative advantages between FDI host and home countries using dyadic analyses. The authors find that bilateral FDI increases the similarity of comparative advantages between a pair, and that this effect is stronger in “South-South” or “North-North” pairs than in other pairs. They also find that bilateral FDI strengthens the spatial correlations of comparative advantage similarity. I think the topic is interesting, but the manuscript has some theoretical and empirical shortcomings (which are listed below). So I am afraid that I cannot recommend it for publication at this moment.
My first concern is that the topic of this manuscript is unrelated to the aim of this journal. It basically has little to do with sustainability. But this should be left for the editor to judge.
Second, the research question as to whether bilateral FDI reduces the comparative advantage gap between two countries is indeed important, but I don’t think the hypotheses are developed well. For example, while elaborating on Hypothesis 1, the authors link FDI to trade, but I don’t know how this linkage to trade is related to the reduction in the gap of comparative advantages between two countries. The authors mention that “two countries with the linkages of FDI should have similar patterns of comparative advantage, thus exporting similar baskets of goods and services” (on page 3). But does this suggest that countries with similar patterns of comparative advantage trade more? If so, the causal relationship discussed in this manuscript is not clear.
Also, I don’t think the four hypotheses in this manuscript are well connected. It is a good idea to look at the heterogeneous effect of FDI (H2), but why do the authors bring in spatial correlation (H3 and H4)? I think each of these could be a topic of an independent paper, but the authors include all of them and the theoretical discussion for each is quite weak. Perhaps the authors can focus only on H1 and H2 (or H1 and H4) and develop the hypotheses better instead of lumping four hypotheses in one paper. Plus, H3 is not really about FDI (which is the key independent variable in this study), so I don’t know it should be listed as a hypothesis.
Third, the biggest concern I have on this manuscript is regarding the endogeneity issue, both theoretically and empirically. As mentioned in the first point, the authors seem to suggest that similar countries trade with (or invest in) each other more. So an endogeneity problem clearly exists here. This is more evident in Hypothesis 2, because countries at the same development level are already similar to each other. It is not surprising, then, that the authors find the effect of FDI is stronger in “South-South” or “North-North” pairs. More importantly, the authors did not deal with this problem in their empirical analysis. Equation (I) on page suggests that all variables are at time t, so how do we know that it is FDI that leads to comparative advantage similarity or the other way around?
Fourth, the title of this manuscript includes “international knowledge diffusion,” but I am not sure whether the empirical analysis has anything to do with international knowledge diffusion. The authors briefly discuss knowledge diffusion in the first two sections, but the dependent variable used in the empirical model is similarity of comparative advantage. What kind of comparative advantage does it measure? Is it about knowledge? If not, then how is this manuscript related to “international knowledge diffusion”? I think at least the authors should provide more information about this variable.
Fifth, the structure of this manuscript also needs to change. Section 4 is unnecessary. It is titled “Data and descriptive statistics,” but data are actually discussed in Section 3. And there is no need to list all maximum and minimum pairs on pages 9-10. This information and the descriptive statistics could be in the appendix or just presented in the table.
Lastly, the writing of this manuscript needs to be improved, as there are many grammar errors. I think it will benefit a lot from professional editing. The authors should also proofread the manuscript more carefully. For example, Table 3 has Chinese characters!
Author Response

(The authors gave the same response as above.)

Round 2
Reviewer 1 Report
The paper was improved according to the recommendations and can be fully considered for publication.
Author Response
Thanks for the reviewer's recognition.
Reviewer 3 Report
I appreciate the authors' efforts to revise the paper and respond to reviewer comments. However, I don't think the authors have properly addressed all issues raised by reviewers. For example, I questioned the endogeneity problem, and the authors simply dismissed it by saying that "our argument does not suggest that countries with similar patterns of comparative advantage trade more, so the endogeneity problem concerned by reviewer should not be a problem." The authors should at least show that endogeneity doesn't exist rather than saying it is not a problem.
Author Response
Dear reviewer,
We thank you very much for the comment on the endogeneity issue. Our understanding is that you may concern a possible reverse causality by the similarity of comparative advantages to bilateral FDI between the source and host countries. We have mulled the reviewer’s concern over, and think that this reverse causality cannot be derived from the existing theories of FDI. As we know, FDI takes two basic forms: horizontal FDI and vertical FDI. Because the horizontal FDI is driven mainly by seeking foreign markets, it is not likely to occur between the countries with a similar comparative advantage. By contrast, the motivation of the vertical FDI is to reduce the production costs through the foreign affiliates. Since the production costs are associated with the relative abundance of resources that determines the comparative advantage at some degree, the vertical FDI could relate to the similarity of comparative advantages. However, if this effect exists, the vertical FDI should be negatively associated with the similarity of comparative advantages instead of a positive relationship. Based upon these reasons, we believe that it is not essential to consider the endogeneity issue that may be from the reverse causality. If you have any other questions, please feel free to let us know.